# Interactions of Nucleosomes with Acidic Patch-Binding Peptides: A Combined Structural Bioinformatics, Molecular Modeling, Fluorescence Polarization, and Single-Molecule FRET Study

**DOI:** 10.3390/ijms242015194

**Published:** 2023-10-14

**Authors:** Pavel D. Oleinikov, Anastasiia S. Fedulova, Grigoriy A. Armeev, Nikita A. Motorin, Lovepreet Singh-Palchevskaia, Anastasiia L. Sivkina, Pavel G. Feskin, Grigory S. Glukhov, Dmitry A. Afonin, Galina A. Komarova, Mikhail P. Kirpichnikov, Vasily M. Studitsky, Alexey V. Feofanov, Alexey K. Shaytan

**Affiliations:** 1Department of Biology, Lomonosov Moscow State University, 119234 Moscow, Russia; 2Laboratory of Structural-Functional Organization of Chromosomes, Institute of Gene Biology, Russian Academy of Sciences, 119334 Moscow, Russia; 3Faculty of Biology, MSU-BIT Shenzhen University, Shenzhen 518172, China; 4Department of Physics, Lomonosov Moscow State University, 119234 Moscow, Russia; 5Shemyakin-Ovchinnikov Institute of Bioorganic Chemistry, Russian Academy of Sciences, 117997 Moscow, Russia; 6Fox Chase Cancer Center, Philadelphia, PA 19111, USA

**Keywords:** nucleosomes, nucleosome-binding peptides, acidic patch, LANA, CENP-C, molecular dynamics simulations, fluorescence polarization, FRET

## Abstract

In eukaryotic organisms, genomic DNA associates with histone proteins to form nucleosomes. Nucleosomes provide a basis for genome compaction, epigenetic markup, and mediate interactions of nuclear proteins with their target DNA loci. A negatively charged (acidic) patch located on the H2A-H2B histone dimer is a characteristic feature of the nucleosomal surface. The acidic patch is a common site in the attachment of various chromatin proteins, including viral ones. Acidic patch-binding peptides present perspective compounds that can be used to modulate chromatin functioning by disrupting interactions of nucleosomes with natural proteins or alternatively targeting artificial moieties to the nucleosomes, which may be beneficial for the development of new therapeutics. In this work, we used several computational and experimental techniques to improve our understanding of how peptides may bind to the acidic patch and what are the consequences of their binding. Through extensive analysis of the PDB database, histone sequence analysis, and molecular dynamic simulations, we elucidated common binding patterns and key interactions that stabilize peptide–nucleosome complexes. Through MD simulations and FRET measurements, we characterized changes in nucleosome dynamics conferred by peptide binding. Using fluorescence polarization and gel electrophoresis, we evaluated the affinity and specificity of the LANA_1-22_ peptide to DNA and nucleosomes. Taken together, our study provides new insights into the different patterns of intermolecular interactions that can be employed by natural and designed peptides to bind to nucleosomes, and the effects of peptide binding on nucleosome dynamics and stability.

## 1. Introduction

Understanding the functioning of eukaryotic genomes including the human genome is one of the key challenges of this century. Eukaryotic genomes reside in the nuclei of eukaryotic cells as a complex structure, called chromatin, consisting of the genomic DNA molecules, nuclear proteins, and RNA molecules. Histones, the most abundant proteins in the nuclei, are indispensable for chromatin organization. Four types of histones (H3, H4, H2A, H2B) form histone octamers that bind to the genomic DNA approximately every 200 bp and form nucleosomes [1]. The core of the nucleosome (the nucleosome core particle, or NCP) is a disc-shaped particle approximately 10 nm in diameter. A total of 145–147 DNA base pairs are wrapped in ~1.7 turns of a left-handed superhelix around the octamer [2,3]. The octamer has a two-fold symmetry axis resulting in the similarity of the top and bottom surfaces of the NCP. Apart from compacting the genomic DNA, nucleosomes serve as integral elements that mediate interactions between chromatin machinery and specific genomic loci [4]. They also provide epigenetic markup for the genome through histone post-translational modifications [5], incorporation of histone variants [6], and specific positioning of the nucleosomes themselves along the genome [7]. Eventually, an interplay between nucleosome dynamics, its modifications, and interactions, provides a basis for genome functioning including gene expression regulation, DNA repair, and replication [4].

When interacting with nucleosomes, chromatin proteins (or chromatin protein complexes) may engage both DNA and histones, including the flexible histone tails. Yet, several distinct regions on the globular surface of nucleosomes are known to provide interaction sites that are engaged repeatedly by different proteins [8]. According to high-throughput pull-down screens, more than 50% of nucleosome interactors engage a characteristic highly countered negatively charged patch on an otherwise predominantly positively charged surface of the histone octamer—the so-called, acidic patch. It was first observed in the original high-resolution NCP X-ray crystal structure as a site promoting nucleosome stacking structure by binding the positively charged H4 histone N-terminal tail of the neighboring nucleosome [2]. The acidic patch is located on the surface of the H2A-H2B histone dimer and is usually defined by eight negatively charged amino acid residues (E56, E61, E64, D90, E91, E92 of H2A and E102, E110 of H2B) [9].

A bunch of 3D atomic resolution structures have now been solved where proteins or their fragments interact with the nucleosome via the acidic patch. Among them, nucleosome remodeling complexes (e.g., RSC [10], BAF [11,12]), chromatin modifiers (e.g., Dot1L [13], PRC1 [14], SAGA [15]), other chromatin protein (e.g., CENP-C [16,17], RCC1 [18], HMGN2 [19], interleukin-33 [20]), anti-nucleosome antibodies (e.g., PL2-6 [21]), and, interestingly, some viral proteins, such as Kaposi’s sarcoma-associated herpesvirus (KSHV) LANA, human cytomegalovirus (hCMV) immediate-early 1 (IE1) protein, and prototype foamy virus (PFV) group antigen (Gag) protein (reviewed in [22]). In these structures, a characteristic feature is seen where an arginine residue side chain (termed an arginine anchor) is inserted into a deeper binding pocket of the acidic patch formed by three H2A acidic residues: E61, D90, and E92 [22,23].

Nucleosome acidic patch-binding peptides derived from such proteins or designed artificially are of practical interest. So far, peptides such as LANA_1-23_ [24], CENP-C_motifs_ from different organisms [25], chromatin-tethering domain (CTD) of IE1 (residues 476–491) [26], the nucleosome-binding motif from HMGN2 protein, H4 histone tail, and artificially designed GMIP1 peptide based on the RCC1–nucleosome complex [27] have been studied. By interacting with the acidic patch, such peptides may potentially be used to alter the epigenetic states of chromatin, block viral replication, or serve as vectors to tether other moieties to nucleosomes. Of these peptides, currently, the LANA peptide is the most studied. Indeed, LANA_1-23_ was found to promote the compaction of nucleosomal arrays [28], used in vitro to block interactions of nucleosomes with chromatin proteins [29], and, recently, to tether chemical catalysts to nucleosome and promote regioselective synthetic post-translational modification of histones in vitro and in living cells [30]. Fang et al. have studied the interactions of the chromatin-tethering domain of IE1 from human cytomegalovirus (hCMV) with the nucleosome core particles [26]. They have found that, similarly to LANA_1-23,_ this peptide impairs the compaction of higher-order chromatin structure, suggesting that in vivo IE1 loosens up the folding of host chromatin during hCMV infections. Kato et al. have studied the interactions of CENP-C_motif_ peptides with centromeric nucleosomes [25]. They have shown that CENP-C_motif_ peptides also bind to canonical nucleosomes, but with lower affinity. The higher affinity of these peptides to centromeric nucleosomes come from an additional interaction with the hydrophobic C-terminal tail of centromeric H3 histone variants. Teles et al. have recently explored the effects of four nucleosome-binding peptides (LANA_1-23_, HMGN2, H4 histone tail, and an artificial peptide identified as GMIP1) both in vitro and in vivo [27]. Their study suggested that the H4 histone tail-derived peptide stands out, is highly specific for targeting the nucleosome, with important effects on the final nucleosome structure, and robust in vivo effects. However, additional experiments are needed to understand if these effects are solely due to acidic patch-binding (or other nonspecific) interactions of the peptide. Previous studies by Chodaparambil et al. suggested that the H4 tail has multiple modes of interaction with the nucleosome, including nonspecific ones [28].

Ideally, nucleosome-binding peptides should have high (yet tunable) affinity and high specificity. The reported data currently suggests that the LANA_1-23_ peptide may bind nucleosomes with an affinity as high as 160 nM [31], although some studies report lower affinity values (~180 nM [32], 250 nM [26], 8 µM [27], 1–9 µM [33]). The CENP-C_motif_ peptides have been reported to bind canonical nucleosomes with an affinity of ~1–6 µM [25], and CTD-IE1 with an affinity of 420 nM [26]. The specificity of peptide binding to the nucleosome with respect to binding to DNA is another factor important for practical applications. The LANA peptide bound to the GST protein was previously reported not to interact with DNA, contrary to the H4 histone tail that was reported to interact with DNA [28].

Despite considerable interest in potential applications of acidic patch-binding peptides, our understanding of their interaction patterns with nucleosomes, their effects on nucleosome dynamics, and the approaches to design peptides with improved/tunable properties are still limited. To address these problems and enhance our understanding of nucleosome-binding peptides, herein, we have applied several computational and experimental approaches. We performed an automated comprehensive analysis of the currently available 3D atomic structures of nucleosome complexes in the Protein Data Bank in order to understand the interaction patterns employed by different peptides and proteins that bind to the acidic patch. We show that apart from a single “anchor” arginine residue (that binds to a specific pocket on the acidic patch surface of the nucleosome), binding patterns of proteins to the acidic patch are variable both in terms of protein sequence and conformation. However, we find that the composition of acidic patch-binding motifs is enriched in arginines and serines at specific positions. To further assess the strength, variability, and dynamics of nucleosome–peptide interactions, we performed molecular dynamic (MD) simulations of the two nucleosome-binding peptides bound to nucleosomes (LANA_1-22_ from Kaposi’s sarcoma-associated herpesvirus and CENP-C_motif_ from *R. norvegicus*). Our results revealed the importance of transient atom–atom interactions for the overall stability of the nucleosome–peptide complex. We also showed that peptide binding produces detectable changes in the conformation of the histone octamer. We further analyzed whether histone sequence variants (due to variation among species or between paralogous genes) may affect the sites of peptide attachment and we found that certain H2A histones of Metamonada, Microsporidians, Lilium, as well as short H2A histone variants of mammals have impaired acidic patches. To address the varying reported Kd values of the LANA peptide binding to NCP, we developed a fluorescent polarization assay and measured the respective Kd values. We confirm that the LANA_1-22_ peptides bind to nucleosomes at submicromolar concentrations, and their binding is specific to nucleosomes but not free DNA. Finally, we employed spFRET measurements to experimentally characterize the effects of peptide binding on nucleosome stability and conformation. We showed that the LANA_1-22_ and CENP-C_motif_ peptide binding makes nucleosomes, on average, more compact.

## 2. Results

### 2.1. Analysis of Nucleosome Complexes from the PDB Database

In order to characterize possible interaction patterns that may be used by peptides and proteins to bind nucleosomes via the acidic patch, we performed an extensive analysis of the PDB database (see Methods Section 4.1). We identified proteins or peptides that were in contact with at least one residue of the acidic patch. As reported earlier, an anchor arginine residue interacting with the negatively charged binding pocket was identified as a hallmark feature in most of the structures [9]. Among 116 selected structures, 90 structures had this characteristic feature; thus, we further focused on analyzing these structures. We have developed a webpage hosted on GitHub with an interactive viewer of different nucleosome–peptide and nucleosome–protein structures, which greatly facilitated the analysis. The viewer is available at https://intbio.org/Oleinikov_et_al_2023/ap_viewer (accessed on 24 September 2023).

#### 2.1.1. Analysis of Nucleosome–Peptide Structures

We first focused our attention on the structures of nucleosomes bound to peptides (less than 50 amino acid residues in length). This provided a smaller subset of six structures for a detailed analysis and allowed for a more straightforward interpretation of the results since only peptides were responsible for nucleosome binding, without potential contributions from other parts of proteins or protein complexes. The results of our analysis are presented in Figure 1.

Major geometrical properties of the nucleosomal surface-mediating peptide interactions may be seen in Figure 1a: (1) a deep negatively charged binding pocket occupied by the anchor arginine residue side chain, (2) a crevice on the surface of the H2A-H2B dimer, and (3) additional basins around the nucleosome pore region. The walls of the crevice are formed by the α1 and αC-helices of the H2B histone and parts of the α2-helix of the H2A histone. The backbone of peptides usually runs along this crevice (Figure 1a, right). In the case of the LANA peptide, its backbone forms a U-shaped structure, where one part of the peptide lies in the crevice and the other folds back on top of it. A similar arrangement is observed for the IE1 peptide. Oppositely, in the case of the CENP-C and KNL2 peptides, the peptide backbone does not fold back after running through the crevice, but continues to run towards the nucleosomal pore region and interacts with this basin.

The analysis of interactions of peptides with nucleosomes was performed to highlight both the contacting residues of the peptides and histones (Figure 1b,c). The anchor arginine stands out almost in all the cases as the residue making the greatest number of atom–atom contacts (see Figure 1b, as much as 37 atom–atom contacts for the IE1-CTD peptide). However, in certain cases, other residues may form a considerable number of contacts too.

On average, we also see that many positively charged residues of the peptides (arginines and lysines) participate in contacts with the nucleosome. It is evident that positively charged amino acids surround the arginine anchor and play a major role in neutralizing charges of the acidic patch.

Interestingly, there is a serine residue in five out of six peptides near the anchor arginine. Additional analysis of hydrogen bonds (see Figure 2) revealed that these serines make hydrogen bonds with the glutamates in many cases, additionally stabilizing the interactions with the acidic patch.

Despite the pronounced polar nature of most interactions, hydrophobic amino acids are also involved in peptide binding. One such example is a tryptophan residue in the CENP-C_motif_ peptide (W17 in CENP-C_motif_, structure with PDB ID 4X23, 28 atom–atom contacts). These contacts are formed with the nucleosomal pore basin, particularly the side chain atoms of R129 of the H3 histone, highlighting the potential importance of other interactions in stabilizing the peptide. In principle, tryptophan residues are known to potentially form π-cation interactions with the arginine residues. This was not the case in the studied structures; however, it may be that they may still be formed transiently during the dynamics. Analysis of histone-interacting residues (Figure 1c) further supports the idea that negatively charged residues of the histone acidic patch are not the only ones involved in interactions with peptides.

While H2AE61, H2AE64, H2AD90, H2AE91, H2AE92, and H2BE102 form the arginine binding pocket (Figure 1a) and are always in contact with the peptides, other negatively charged amino acids are located outside of the pocket and form contacts with arginines and lysines (H2AE56, H2BE110) peptides. It is important that not only acidic residues are involved in binding, for most peptides contacts of histidine (H2BH106) located near the arginine anchor pocket are seen, as well as contacts of glutamine (H2BQ44) with polar amino acids of peptides. There are also some nonpolar amino acids that interact with peptides (H2BV45, H2BL103), but they are located near other polar residues.

Even when analyzing a small number of peptides, as well as comparing them to other similar proteins, it is evident that a highly similar sequence does not necessarily lead to the formation of similar contacts. For example, the PL2-6 antibody fragment contains motifs that are present in CENP-C, but they do not participate in contact formation (see framed regions in Figure 1b). The CENP-C_motif_ itself can adopt different conformations in the nucleosome pore region (see Figure 1a, right). At the same time, the GAG protein segment has a highly similar interaction profile to the one observed for CENP-C_motif_, but a has significant difference in its sequence.

#### 2.1.2. Analysis of Nucleosome–Protein Structures

To further elucidate potential interaction pattern characteristics of the nucleosome acidic patch, we extended our analysis not only to peptides, but to proteins and protein complexes of any size interacting with the acidic patch. Among 90 PDB structures, we have selected 39 structures with unique sequences of the interacting protein/peptide motifs. Figure 2 summarizes the results of our analysis.

We first aimed at characterizing the possible conformational arrangements of the protein motifs interacting with the acidic patch. In Figure 2a, in the top left panel, one can see the superposition of all the motifs and different conformations of the anchor arginines present in the structures. A certain degree of conformational flexibility is observed for the anchor arginine. Depending on the conformation, position, and arrangement of the residues surrounding the anchor arginine, we were able to manually group structures into seven classes (Figure 2a). A few structures did not have considerable interactions beyond the interaction with the anchor arginine pocket (“pocket only” class). Other structures belonged to two types of classes differing in the direction of the peptide motif with respect to the crevice—down-up (DU, the C-end of the peptide is closer to the nucleosome center) or up-down (UD, the N-end of the peptide is closer to the nucleosome center). It is interesting that, despite the differences in the direction of the protein chain, the anchor arginine occupies a very similar position. The categories were further varied by the conformation of motifs and parts of the nucleosome surface engaged by the motif for interactions. One particular class is the DU 2 class, where α-helices lie along the crevice. The most characteristic feature of the structures of this class is the periodicity in the arrangement of positively charged amino acids (Figure 2b). Consequently, arginines and lysines are exposed on one side of the α-helix and interact effectively with the acidic patch. In cluster UD1, proteins interact only with a portion of the surface below the arginine pocket (e.g., LANA). Class UD3 contains structures in which proteins interact not only with the acidic patch but also with H3 histone near the nucleosomal pore (e.g., CENP-C_motif_). Class UD4 presents structures with an alternative binding mode, where the protein chain lies along the H2A histone and runs towards H4. All the structures presented in these classes are available at https://intbio.org/Oleinikov_et_al_2023/ap_viewer (accessed on 24 September 2023).

Despite noticeable differences in the type of protein interactions with the surface of the acidic patch, the sequences do not convey to any strict pattern defining one or another class (Figure 2b). Probably, the differences in this case are determined by interactions with other protein sites distant from the arginine anchor. However, many arginine and lysine residues are found in all sequences (Figure 2b,e), which is generally expected for proteins that neutralize acidic residues on the nucleosome. It is noteworthy that most structures (except DU2 class which contains α-helices) contain serine residues near the arginine anchor. In addition, nonpolar leucine, valine, and proline are often found in interacting sites. Negatively charged amino acids are also found in a significant number of proteins (Figure 2d). Interestingly, the distribution of amino acids is not random (Figure 2c). It can be seen that positively charged amino acids are predominantly found in the region of the arginine pocket, near H2AE56, and in the region above the nucleosomal pore. Serine residues are located between them, apparently additionally stabilizing the protein in a highly polar environment. Approximately every third serine residue is involved in hydrogen bonding with H2AE61 and H2A64. The sequence logo (Figure 2e) also shows that serine residues are often present between islands of arginines ±3 amino acids relative to the anchor arginine. At the same time, negatively charged amino acids are often bound to the positively charged H2B surface on the right side of the binding crevice (Figure 2c). Thus, all detected proteins recognize a complex footprint of amino acid characteristics that allows for strong binding to the surface through charge and polar interactions. Similar to the interactions with peptides, when analyzing the whole set of proteins, it can be seen that residues of H2AE61 and H2AE64 almost always participate in interactions, but H2AE64 forms less contacts (on average). Interestingly, residue of H2BH106 is more frequently involved in interactions with proteins than negatively charged neighboring glutamate residues (Figure 2f).

### 2.2. Molecular Dynamics Simulations of LANA_1-22_ and CENP-C_motif_ Bound Nucleosomes

#### 2.2.1. Characterization of Binding Dynamics and Stability of Contacts

To further understand the dynamical patterns and effects of peptide nucleosome binding, we performed MD simulations of NCP-bound structures for two peptides: LANA_1-22_ peptide—a fragment of LANA protein from Kaposi’s sarcoma-associated herpesvirus—and CENP-C_motif_—a fragment of CENP-C protein (amino acids 710–734) from *R. norvegicus*. Prepared systems contained *H. sapiens* histones with truncated flexible histone tails, α-satellite DNA, and one or two copies of LANA_1-22_ or CENP-C_motif_, respectively (see details in Methods Section 4.2 and Appendix A). Obtained MD trajectories showed that full-length peptides remained bound to their respective NCPs (see Figure 3a,b, Appendix A, and interactive trajectory preview at https://intbio.org/Oleinikov_et_al_2023/ (accessed on 24 September 2023)). The central part of peptides which includes the anchor arginine and neighboring residues remained stably bound to the nucleosome core, while the ends of the peptides manifested significantly higher flexibility. Peptides’ flexible tails had time-averaged Cα-atom position deviations of up to 1 nm (Appendix A) and formed a remarkably lower number of atom–atom contacts with the NCP than the central parts of the peptides (Figure 3c,d). The location of these flexible tails correlated with unresolved peptide residues in experimental structures (PDB IDs 1ZLA and 4X23). The relative orientation and conformation of the C-end of the CENP-C_motif_ changed in MD simulations with respect to the ones observed in the experimental X-ray structure. This change is likely due to the fact that crystal contacts between CENP-C_motif_ and its copy in the neighboring unit of the crystal lattice in the X-ray structure are not present in the MD system. The central regions of the peptides, including anchor arginines, established relatively stable conformations and manifested restricted flexibility (see insets in Figure 3a,b). In our MD simulations, anchor arginines (R8 and R9 for CENP-C_motif_ and LANA_1-22_, respectively) formed contacts with particular acidic patch residues (E61 and D90 H2A) in more than 90% of MD frames. Overall, anchor arginines provided the highest average number of atom–atom contacts with the NCP (Figure 3c,d). Interestingly, in one simulation repeat on one side of the NCP, we observed the reorganization of the cluster of contacts formed by R8 and R10 of the CENP-C_motif_ with E61, E90 of H2A, and E102 of H2B. R8 lost contacts with E61 but gained contacts with E102, simultaneously, and R10 substituted R8 in contacting E61 (see Figure 3d, molecular details in Appendix A). It caused a small displacement of the anchor R8, but it still made strong interactions with the acidic patch residues around the original site of its interaction. Peptide residues around anchor arginines showed extensive contacts with other residues of the acidic patch and nearby histone residues with varying probability (Figure 3c,d). Some of these contacts were stable contacts (found in more than 70% of MD frames). Particularly, for LANA_1-22_,we find additional stable contacts between R7 and H2BE110, H2BH106, M6, and H2BE110, and R12 and H2BV45, H2AE64, T14, and H2BQ44. Additionally, for CENP-C_motif_, stable contacts were observed between R10 and H2AE61, and L11 and H2AE91. Another important CENP-C_motif_-specific interaction feature is the interaction with H3 α3-helix residues (Figure 3d,e, Appendix A). This interaction is based on the hydrophobic interaction of Y16 and W17 of CENP-C_motif_ residues with a hydrophobic cavity formed by the backbone and side chain atoms of H3 α3-helix residues around H3L126 (Figure 3e, Appendix A). Contacts between CENP-C_motif_ W17 and the H3 α3-helix were stable in 3 out of 4 simulation replicates (Appendix A). In the fourth replicate, CENP-C_motif_ formed a compact structure using interactions between its own residues (Appendix A). However, even such peptide confirmation retained contacts with the H3 α3-helix (through polar interactions with H3R128). Thus, CENP-C_motif_ simultaneously forms stable contacts with the H2A-H2B dimer (including contacts formed by anchor R8 and acidic patch) and H3 histones. Apart from stable contacts, LANA_1-22_ and CENP-C_motif_ form a lot of other contacts with the NCP (Figure 3c,d, Appendix A). The average number of atom–atom contacts was 125 ± 94 and 80 ± 20 for CENP-C_motif_ and LANA_1-22_, respectively. Most of these contacts were transient (were present in less than 70% of MD frames). Transient contacts included interactions with the acidic patch. LANA_1-22_ also formed transient contacts with the DNA (using R20 and S22, see Figure 3c). LANA–DNA contacts involved DNA base pairs located in a wide DNA segment (from −54 to −43 residues of chain J and from 46 to 48 residues of chain I). However, our simulations used NCPs with truncated histone tails. MD simulations of NCP with full-length histone tails showed that these DNA sites interact with the H2A C-tail; thus, an additional competition with this tail for binding to the DNA may take place [34,35]. Next, we tried to understand the contribution of transient contacts formed by flexible LANA_1-22_ tails in the formation of a stable NCP–peptide complex. We defined tails as residues that are not resolved in the crystal structure (PDB ID 1ZLA) and showed high RMSF values (Appendix A) in our MD simulations and formed only ~20% of atom–atom NCP–peptide contacts (average number of tails’ contacts was 13.7 ± 2.9). The system of LANA_4-17_ without flexible peptide tails was simulated (Appendix A). In the MD simulations of LANA_4-17_–NCP, we observed that LANA_4-17_ lost stable interaction with the NCP and bound to the DNA. This suggests that flexible LANA_1-22_ tails and their transient interactions with the NCP are important for complex stability. Taken together, MD simulations showed that LANA_1-22_ and CENP-C_motif_ form stable contacts with the NCP-embedded H2A-H2B dimer through the anchor arginines and surrounding residues. CENP-C_motif_ has an additional site for stable hydrophobic contact with the H3 α3-helices. Most of the NCP–peptide contacts are transient. Nonetheless, such transient contacts are important for the stability of the NCP–peptide complex.

#### 2.2.2. Effects on Nucleosome Dynamics and Conformation

Next, we aimed to analyze the effects of peptide binding on nucleosome conformation. Previously, using MD simulations, we have shown the presence of inner nucleosome core plasticity, which plays an important role in nucleosome functional dynamics [34,36]. For instance, the bending of H2A and H2B α2-helices was shown to be associated with DNA sliding and DNA unwrapping dynamics within the nucleosome. Moreover, we showed that variant histones may confer varying degree of bendability to the H2A-H2B dimers and in this way affect the dynamical properties of the nucleosome [36]. Nucleosome-binding peptides interact with the nucleosome acidic patch, which includes some residues of the H2A α2-helix (Figure 4a). We hypothesized that bending dynamics may be affected by the binding of peptides. Indeed, in spite of the overall similar geometry of the NCP with and without peptides (Appendix A), we observed some differences in the bending of the H2A-H2B dimer upon LANA_1-22_ binding (Figure 4a,b, Appendix A). Here, we used the α2-α2 angle as a measure of the H2A-H2B dimer bending (see Methods Section 4.2 for definition, schematically represented in Figure 4a). Upon binding, the angle changed by ~2–2.5 degrees in the direction opposite to the NCP center. These changes are similar to those happening to the dimer bending upon DNA unwrapping [36]. LANA_1-22_ binding only influenced the conformation of the interacting dimer; the conformation of the LANA_1-22_-free dimer on the other side of the NCP in the same trajectories was not altered (Appendix A). In the first trajectory, we observed altered bending of the LANA_1-22_-free dimer; however, this bending was likely due to the DNA unwrapping (see interactive trajectory preview https://intbio.org/Oleinikov_et_al_2023/NCP_tt_lana3_trj_preview (accessed on 24 September 2023)). We did not observe significant changes in the H2A-H2B dimer bending dynamics upon CENP-C_motif_ binding (Appendix A). Bending of the H2A-H2B α2-helices, which is associated with LANA_1-22_ binding, lead to the increase in the distances between αC-helices of the two H2A histone molecules in the NCP and, to a lesser extent, between H2A α3-helices (Appendix A). Interestingly, even without altering H2A-H2B dimer bending, CENP-C_motif_ binding also lead to the increase of the distance between the H2A αC-αC-helices (Figure 4c,d) and H2A α3-α3-helices (Appendix A). Distance between the H2A αC-αC helices was increased by ~2.5 Å in LANA_1-22_–NCP simulations and 3 Å in CENP-C_motif_–NCP simulations (Appendix A). How does CENP-C_motif_ affect the H2A αC-αC-helices’ distance without bending the H2A-H2B dimer? The probable answer is the bivalent nature of CENP-C_motif_ binding (simultaneous stable interaction with the H2A-H2B dimer and H3 α3-helices) (Figure 4c,d). NCP simulations without peptides show that there are contacts between the H2A αC-helix and H3 α3-helix. CENP-C_motif_ binding may decrease the number of H2A-H3 contacts (Appendix A). Thus, submicrosecond MD simulations of LANA_1-22_ and CENP-C_motif_-bound NCP have shown that peptides’ binding impacts nucleosome core geometry. In particular, upon binding, we observed an increase in the distance between the H2A αC-helices. Due to different binding modes, along with the increase of αC-αC separation, peptides affected the H2A-H2B dimer bending (LANA_1-22_) or the number of H2A-H3 contacts (CENP-C_motif_).

### 2.3. Histone Sequence Variants and Potential Effects on Peptide–Nucleosome Interactions

Histones are among the most conserved proteins in evolution; however, families of replication-independent histone variant genes that may differ considerably in their amino acid sequence from the canonical histones are known to play important roles in genome regulation [6]. Moreover, small changes in the sequence of canonical histones between species or within the paralogous genes in the genome of a specific species may differentially affect nucleosome stability and chromatin functioning [37]. Having characterized the important contact sites on the histone octamer surface engaged during peptide binding in Section 2.1 and Section 2.2, we next set to analyze if those sites are perturbed due to sequence variations in H2A histones across histone variants and species. We compiled a set of 270 diverse histone H2A sequences based on the curated part of the HistoneDB database supplemented with some newly characterized sequences (see Methods Section 4.3). Multiple sequence alignments were built to specifically analyze changes in the histone sequences at the following positions corresponding to the canonical H2A histone (the histone used in MD simulations): H2AE56, H2AE57, H2AE61, H2AE64, H2AD90, H2AD91, and H2AD92. Appendix A provide an overview of the obtained alignments, while Figure 5 presents examples of most notable changes in different species and variants.

At first, we focused on the human genome containing 29 known H2A histone coding genes [37]. Figure 5a demonstrates that only two human histone variants, H2A.B and H2A.P, have variations in the acidic patch region which should be important for LANA–peptide binding (see also Appendix A). Further, we analyzed if there are more significant changes in H2A histone variants in other species. Across all animals (Metazoa), we only observed H2A.B, H2A.L, H2A.P, and H2A.Q sequences with variations in the major contact sites on the surface of the histone octamer involved in peptide binding. These four variants belong to a class of short H2A histones found in placental (eutherian) mammals and lead to loosely packed chromatin due to a shortened C-terminal tail and a weak acidic patch [38,39,40]. Sequences of short H2A of other animals, similarly to human H2A.B and H2A.P histones, lack negatively charged amino acids in most positions in the contact sites on the histone octamer surface important for nucleosome functioning (Figure 5a and Appendix A). Additionally, amino acid changes in short H2As are noticeable not only between variants, but also at the level of different species. For example, the E61K substitution in H2A.B histones is common in Primates, including *H. sapience*, and Carnivora. At the same time, in the mouse orthologues of the human genes encoding H2A.B, the E61R substitution is observed that characterizes Rodentia H2A.Bs (Figure 5a and Appendix A).

Analysis of plant (Viridiplantae) variants revealed substitutions Y56N, E64D, and E91V (Figure 5b and Appendix A) in a gH2A variant that is a gamete-specific variant found in the genus Lilium [41,42]. Interestingly, genes of short H2As and gH2As are specifically expressed in male gametic cells, suggesting that special variations in the acidic patch region may play an important role in gametogenesis.

Furthermore, some significant changes were observed in unicellular eukaryotic parasites, particularly in *G. lamblia*. As previously shown, the *G. lamblia* nucleosome cannot bind the LANA–peptide due to differences in the acidic patch region [16]. Comparing other H2A histone sequences of Metamonada suggests that substitutions of E64R and E91K are typical for various Giardia species (Figure 5c and Appendix A). Among other Metamonada H2As, there are some sequences with changes in the important contact sites on the histone octamer surface involved in LANA–peptide binding. For instance, *S. salmonicida* H2As are characterized by substitutions Y57S, E64K and E91K, and *A. ignava* H2As–E91R (Figure 5c and Appendix A).

Another finding in our analysis was that there are some Microsporidian H2A histone sequences that have different variations in positions E56, E64, and E91 (Figure 5d and Appendix A).

It is important to note that there are positively charged amino acids positioned within the acidic patch region and other contact sites significant for LANA–peptide binding in Mammalian short H2As (E61K or E61R substitution in most of H2A.Bs, E56K, and E64K are found in some H2A.Qs) and Metamonada H2As (e.g., E64R and E91K in Giardia H2As). This circumstance will affect nucleosome stability and binding of the positively charged peptides. In addition, we observed some variations with hydrophobic and polar uncharged amino acids. For instance, alanine and leucine are common for H2A.Bs in positions E91 and E92, respectively, glutamine and isoleucine are typical for Primates H2A.Qs in positions E56 and E57, respectively, and valine is typical for gH2As in position E91.

### 2.4. Experimental Estimation of Peptide-Nucleosome Interactions

To experimentally evaluate the specificity and strength of interactions between acidic patch-binding peptides, we employed electrophoretic assays and fluorescence polarization (FP) assays. The LANA_1-22_ peptide was used for these studies, which was synthesized with a fluorescent FAM label attached to its N-terminus (see Methods Section 4.7). For both gel electrophoresis and FP measurements, non-fluorescently labeled nucleosomes were reconstituted using 147 bp Widom 603 DNA and histone octamers (see Methods, Section 4.4 and Section 4.5, Appendix A).

To confirm LANA_1-22_ binding to nucleosomes and to estimate its specificity in binding to nucleosomes versus binding to the free nucleosomal DNA, samples were incubated with a high 25 µM concentration of FAM-labeled LANA_1-22_ and subjected to native PAGE with a 6% gel. The scans of the gel showing the position of FAM-labeled LANA_1-22_ and DNA/nucleosomes stained with SYBR Gold are shown in Figure 6a,b. It can be seen that only the bands corresponding to the position of nucleosome, and not the DNA, showed characteristic fluorescence in the FAM channel. This confirms the binding of LANA_1-22_ to the nucleosomes and the absence of its binding to the free DNA. A feature of the electrophoretic assay is the smeared presence of the FAM signal in the gel lane above the position of the nucleosomal band. This suggests that, during the gel electrophoresis, some peptides dissociate from nucleosomes and are left along the path of the band while it migrates in the gel. This effect likely impedes the use of electrophoretic assays for quantitative estimation of dissociation constants. 

Hence, to quantitatively estimate the dissociation constant for the LANA_1-22_–nucleosome complex, we have resorted to a fluorescence polarization assay. The schematic diagram of the experiment is shown in Figure 6c. FAM-labeled LANA_1-22_ peptides at a concentration of 100 nM were mixed with varying concentration of unlabeled nucleosomes. The FAM label was excited with polarized light; the binding of the peptide to the nucleosome was measured by the increase of the polarized component of the emitted light (see Methods, Section 4.9). The measured polarization anisotropy values correspond to the fraction of the bound peptide and were fitted using a corresponding binding model (see Methods, Section 4.9). The estimated value of Kd was 0.4 ± 0.09 µM (see Figure 6d). We consider it as the upper bound for the Kd value, since the effective concentration of nucleosomes during the measurement may be somewhat smaller due to the nonspecific adsorption of nucleosomes on the wells of the plate containing the samples during FP measurements.

### 2.5. Estimating Effects of Peptide Binding on Nucleosome Geometry Using Single-Molecule FRET

To experimentally evaluate potential changes in the geometry and dynamics of mononucleosomes upon peptide binding, we have employed spFRET measurements. To this end, non-labeled LANA_1-22_ and CENP-C_motif_ peptides were obtained with chemical synthesis (see Methods, Section 4.7, Appendix A), and fluorescently labeled mononucleosomes were assembled on a 603 high-affinity nucleosome positioning DNA sequence (147 bp in length). The Cy3 and Cy5 fluorescent labels were attached to the two different DNA gyres at positions located approximately half a turn of the nucleosomal superhelix from the dyad in different directions (see Methods, Section 4.6, positions 35 and 112 bp when numbered from the start of the nucleosomal DNA, or +38 and −39 if numbering starts from the dyad; Appendix A shows the results of native PAGE for the assembled nucleosomes). The labels were attached to thymine bases that point outwards in assembled nucleosomes to avoid interference with DNA–histone contacts (see Figure 7a) [43,44]. As demonstrated before, this experimental setup allows to measure efficiently the changes in the distance between the DNA gyres within mononucleosomes [45,46].

For every nucleosome diffusing through the focal volume of the microscope, FRET efficiency (which is related to the distance between the labels) was characterized by calculating the proximity ratio (E_PR_) (see Methods, Section 4.10). The relative frequency distribution of proximity ratios for mononucleosomes in the presence and absence of the acidic patch-binding peptides is shown in Figure 7b.

In all measurements, we observed a distribution of E_PR_ values with a single maximum, confirming the presence of a FRET effect between the labels and hence the close proximity of the labels and DNA gyres to one another (Figure 7b). The broad shape of the distribution is likely due to several factors usually affecting single-molecule measurements: the presence of background noise, dynamics of the labels (in part due to the flexible linkers used to attach them), and the dynamics of the nucleosome core particle itself. Different modes of nucleosome dynamics such as DNA spontaneous unwrapping/breathing, or even histone dimer exchange are known to be present in nucleosomes [4,47] and likely contribute to the broadening of the distribution of the E_PR_ values. 

In the presence of the LANA_1-22_ peptide and the CENP-C_motif_ peptide, the spFRET analysis of mononucleosomes revealed a shift of the E_PR_ value distribution towards higher E_PR_ values (Figure 7b). This result may be interpreted as the decrease of both the most probable distance and the average distance between the fluorescent labels (and hence the DNA gyres) in the population of the mononucleosomes. Such changes may be related both to the reduction of local fluctuations (e.g., gaping motion between the DNA gyres) or to the suppression of larger-scale conformational fluctuations (e.g., DNA unwrapping). This may be described as a stabilization of the nucleosome, where nucleosomes become on average more compact. As seen from Figure 7b, the effects of CENP-C_motif_ on the average nucleosome compactness were higher than that of LANA_1-22_.

Interestingly, previous studies reported similar effects due to the binding of the CENP-C central domain (which incorporates the CENP-C_motif_) on the structure of centromere nucleosomes (which harbor centromeric H3 histone variant CENP-A instead of canonical H3 histone) [48,49]. FRET studies have indicated that centromeric nucleosomes adopted a more compact shape upon CENP-C central domain binding reminiscent of the shape observed for canonical nucleosomes. Our results suggest that peptide binding may have a stabilizing effect on canonical nucleosomes too.

## 3. Discussion

Compounds targeting the acidic patch of the nucleosome have been explored as potential anticancer agents (causing cytotoxicity through chromatin condensation) [50], antiviral agents (by blocking viral episomal tethering to nucleosomes) [32], delivery vehicles for site-specific nucleosome modifications [51]. Hence, the development of efficient acidic patch binders and modulators of such binding between nucleosome and native chromatin proteins presents considerable interest. In this work, we have characterized binding patterns and dynamical effects of acidic patch-binding peptides via a number of computational and experimental methods. Below, we discuss our findings in the view of the above-mentioned rationale.

Through comprehensive analysis of PDB structures and MD simulations, we have characterized typical interaction patterns of nucleosomes with their acidic patch-binding peptides and proteins. As suggested previously [9], we confirm that the interaction of an anchor arginine residue of a peptide with a negatively charged pocket on the surface of the octamer enclosed by four acidic residues (H2A E61, E90, E91 H2B E102) is a dominant feature found in 90 of 116 structures of nucleosome complexes with different chromatin proteins. Our MD simulations data indicate that stable contacts between peptides and NCP are formed not only by the anchor arginine but also by a segment of 3–5 amino acid residues to the left and to the right of the anchor arginine in the sequence. This segment around 10 amino acid residues in length forms the main interaction with the nucleosome. However, this region alone is not likely enough to stabilize the peptide in place. As shown in our MD simulations, the removal of the terminal parts of the LANA_1-22_ peptide that form only transient contacts with NCP leads, nonetheless, to peptide destabilization and detachment of the central region including anchor arginine. Previous analyses [20,52] suggested that there might be some sequence motifs in the vicinity of the anchor arginine that determine the peptide binding. However, as shown, for example, for CENP-C and PL2-6 fragments, even high sequence similarity does not guarantee a common interaction pattern. Our analysis reveals a more complicated and polymorphic picture. There are certain preferences in sequence composition (preponderance of arginines and serines), and clearly similar interaction patterns engaged by peptides or protein fragments (electrostatic interactions with acidic patch glutamates and aspartates; hydrogen bonds with serines; ion pairs of opposite nature between Asp and Glu in peptides to Arg and Lys of histones;); however, they may be realized by a multitude of different sequence arrangements (both in sequence space and in conformational space). Moreover, peptides and proteins may engage different regions of the nucleosome surface adjacent to the acidic patch to further stabilize their interaction. The region around the nucleosome pore is one of such important regions that is engaged by some peptides (e.g., hydrophobic interactions with H3 for CENP-C_motif_). Minor interactions with the nucleosomal DNA may also contribute to the stability of peptide–nucleosome complex. These findings suggest that there is ample potential to design new artificial nucleosome-binding peptides that would synergistically engage distinct interaction sites within and around the acidic patch of the nucleosome. Rational design of peptide binders have, so far, been computationally challenging, but very recent advances in artificial intelligence algorithms promise considerable prospects [53]. The need to simultaneously engage different sites on the nucleosome is further supported by the attempts to design synthetic compounds that would displace the LANA peptide from the nucleosome or bind to the acidic patch. Namely, a high-throughput screening study of ~350,000 chemical compounds did not find compounds that may specifically displace LANA from the nucleosome surface [32], while another study was able to design larger bispecific compounds interacting with the acidic patch [50].

So far, one study has reported a development of a new nucleosome-binding peptide (named GIMP1) based on a fragment of RCC1 [27]. The GIMP1 peptide had an additional domain that bound DNA to increase its nucleosome affinity. However, likely because of this fact, it showed very different biophysical and biological effects compared to the LANA peptide, which binds mainly to the acidic patch: it showed nonspecific DNA interactions, induced nucleosome precipitation, and did not affect chromatin compaction and viability of tumor cells, unlike LANA. In this study, via MD simulations, we find that the nucleosome-bound LANA_1-22_ peptide has minor interactions with the nucleosomal DNA, and we experimentally confirm that LANA_1-22_ does not bind to free DNA outside of the nucleosome context. The ability of peptides to bind histones specifically and to avoid extensive nonspecific interactions with DNA is likely an important property influencing their potential practical applications. 

The different behavior of specific nucleosome-binding peptides versus peptides that have nonspecific DNA-binding modes was previously shown in classical studies by Chodaparambil et al. in which it was found that LANA peptides induce chromatin fiber self-association in a saturable way, while H4 tail-derived peptides (which bind to the acidic patch, but also bind to the free DNA) continue to promote nucleosomal array self-association when added at increasing concentrations [28]. From that study and [50], we know that in vivo acidic patch-binding compounds induce aberrant chromatin condensation, likely by shielding the repulsive electrostatic interactions of the acidic patch. Whether the addition of DNA-binding domains to the peptides may enhance these effects is unknown, but the nonspecific binding will likely complicate their potential usefulness as therapeutic agents. Hence, we propose that design of better peptide binders should focus on specific stereochemical targeting of molecular interaction fingerprints at the surface of the histone core. 

Peptide binding affinity is an important factor for practical applications of nucleosome-binding peptides. For instance, development of antiviral agents blocking viral DNA tethering to nucleosomes would require new peptide inhibitors with binding affinities greater than those of the viral peptides [32]. Usual interaction affinities of histone tails with their chromatin reader domains are in the micromolar range [54]; hence, this may be considered as an approximate figure for the potential application of peptides as epigenetic agents. In this respect, understanding the quantitative values of the binding affinities of the acidic patch-binding peptides and their dependence on the nucleosome environment is important. For the most studied nucleosome-binding peptide—the LANA peptide—conflicting binding affinity estimates have been reported. Studies based on fluorescence polarization and thermophoresis assays have reported Kd values around 200 nM, while studies using gel electrophoresis to detect and quantify LANA-bound nucleosome reported estimates in the range of 1–10 µM [16,27,33]. We revisited this question in this work using a fluorescence polarization assay and reported a binding constant of around 400 nM ± 0.1 nM or lower, suggesting that the LANA nucleosome-binding constant may indeed be in the submicromolar range, at least for the binding buffer used in this study. Our analysis of nucleosome LANA binding through PAGE suggests a potential explanation for the earlier reported discrepancies. We see that, during the migration of nucleosomes, the positively charged peptide dissociates from nucleosomes, leaving a characteristic smear on the gel lane above the position of the nucleosome band. This fact likely diminishes the actual concentration of the peptide in the nucleosome band and may lead to the underestimation of the peptide-binding affinity. For FP assays, buffer conditions are usually optimized to allow for a better signal-to-noise ratio and to avoid aggregation and adsorption of the molecules [55]. This and previous studies have used buffers of low ionic strength (2–10 mM NaCl) and surfactants (Triton X-100, NP-40) to suppress aggregation and absorption. To what extent these conditions affect Kd measurements remains to be explored in detail.

We have shown that binding of LANA_1-22_ and CENP-C_motif_ peptides may affect average nucleosome geometry. FRET studies support the idea that nucleosomes become, on average, more compact upon peptide binding, while MD simulations showed that small, yet detectable changes occur in the geometry of the histone core. Current consensus suggests that nucleosome functional dynamics consist of large-scale motions such as DNA unwrapping, and more subtle motions, such as nucleosome octamer plasticity [34]. It is known that when protein domains bind to nucleosomes, they may increase their overall stability as judged by the resolution of the cryo-EM maps [56], or induce small structural transition in the nucleosome core of centromeric nucleosomes harboring CENP-A variant, including DNA gyre sliding [48,49]. MD simulations of canonical nucleosomes suggest that histone octamer experiences constant dynamical fluctuations with amplitude of certain α-helices bending of around 7 Å, which may be coupled to the dynamics of DNA twist defects within the nucleosome [34]. Hence, available FRET data may be likely consistent both with the decrease of the DNA inter gyre distance due to suppression of local fluctuations (gapping motions, DNA twist effects sliding) or with stabilization of nucleosomes during larger scale dynamics, e.g., suppression of DNA unwrapping.

One mechanism contributing to nucleosome stabilization by the acidic patch-binding peptides may have an electrostatic nature. The reduction of the acidic patch negative charge with peptide binding effectively increases the positive charge of the histone core and thus strengthens the interactions with DNA. Our results also showed that CENP-C_motif_ makes nucleosomes more compact than LANA_1-22_, on average. We hypothesize that the increase in nucleosomal stability conferred by CENP-C_motif_ may be due to two facts. First, the overall positive charge of the CENP-C_motif_ is higher than for LANA_1-22_; thus, the effective increase in the charge of the octamer will be higher for CENP-C_motif_ than for LANA_1-22_. Second, CENP-C_motif_ has additional contacts with the basin surrounding the nucleosomal pore; thus, it may effectively tether H2A-H2B dimer to the H3-H4 tetramer. Previous high-precision FRET studies have revealed that the interface between the H2A-H2B dimers and the (H3–H4)_2_ tetramer may spontaneously open asymmetrically by an angle of ≈ 20° [57], and the tethering of H2A-H2B dimer with CENP-C_motif_ may suppress this effect. 

Considering that nucleosomal DNA dynamics contribute to significant conformational variability of the chromatin fiber [34], it would be reasonable to assume that the LANA_1-22_’s stabilizing effect to the nucleosome structure due to the suppression of local fluctuations may contribute to fiber–fiber self-association and, therefore, chromatin compaction, which was previously shown using the model system that mimics fiber–fiber interactions in cellular chromatin [58] and in vivo [28]. It is also known that nucleosome stability affects transcription, for instance, due to the nucleosomal barrier for RNA polymerase II [59,60]. Given that the full-length LANA proteins have been previously shown to affect transcription in vivo [61,62], investigating the contribution of the LANA-induced nucleosome stabilization to such effects might be an interesting further direction for investigation.

Recently, the catalyst system PEG-LANA-DSSMe was designed to promote regioselective synthetic histone acetylation at H2BK120 in living cells; hence, the H2B ubiquitination was suppressed [63]. The stabilizing effect of LANA_1-22_ and other peptides may be considered as another advantage of using such peptides in practical applications.

An interesting finding in our MD simulations also shows that CENP-C_motif_ binding affects the distance between the two αC-helices within the histone octamers and affects contacts of H3 and H2A histones near the nucleosomal pore. Due to the two-fold symmetry of the histone octamer, this means that the two CENP-C_motif_ molecules binding nucleosomes from different sides may experience a degree of allosteric communication. In vivo, two CENP-C proteins engage nucleosomes from both sides, and they further associate through a dimerization domain [64]. The degree of this communication requires further investigation. However, there is currently evidence that moieties binding the two sides of the nucleosomes might feel the presence of each other. Certain studies have shown that, for instance, HMGN1/2 and the Fab fragment of the PL2-6 antibody, which both bind the acidic patch of the nucleosome, show cooperative binding behavior in higher ionic strength buffers [52,65]. Other studies of SNF2h remodelers binding to nucleosomes (which also engage the acidic patch) suggested that the binding of the remodeler from one side may induce disorder in the acidic patch on the other side of the nucleosome and thus inhibit the binding of the second copy of the remodeler, which would try to slide the nucleosome in the other direction [66].

## 4. Materials and Methods

### 4.1. PDB Structure Analysis

Using the nucleosome database (NuclDB, http://nucldb.intbio.org (accessed on 24 September 2023), [67]), we have aligned and extracted all the structures containing nucleosomes bound to other interactors (proteins or peptides) from the PDB database, available as of 08/24/2023. All structures were superimposed with the reference structure (PDB ID 1KX5) by sequence-structure alignments using in-house Python scripts. We selected 116 structures that contained proteins interacting with the acidic patch by looking for non-histone atoms within 4 Å from the acidic patch residues. As structures contained different histone variants and residue numbering schemes, we extracted acidic patch residue numbers from sequence alignments with canonical *X. laevis* histones. We selected 90 structures that contained an arginine residue within 5 Å from both E61 and D92 residues of H2A (we used numbering for canonical H2A of *X. laevis*). We marked that residue as arginine anchor and used it to create sequence alignments and used it as a reference point for further analysis of interacting protein structures. Since there are two acidic patches for each nucleosome, we detected the interacting side and aligned all structures to match the interacting side. All analysis was performed with MDAnalysis version 2.6.1 [68] library. Visualization was performed with Chimera version 1.15 (University of California, San Francisco, CA, USA), VMD version 1.9.3 (University of Illinois, Urbana-Champaign, IL, USA), and NGLview version 3 [69,70,71]. For all interacting proteins, we calculated the per-residue atom–atom contact profiles with 4 Å threshold; hydrogen bonding was calculated with HBPLUS version 3.06 [72]. We also calculated such contact profiles for histones, as structures contained different histone variants; contact profiles were mapped to canonical histone sequences by pairwise alignment. Sequence alignments and logos were visualized using custom Python scripts.

### 4.2. MD Simulations and Analysis

All-atom MD simulations of NCP with LANA_1-22_ and CENP-C_motif_ were performed in several replicates. An overview of simulated systems and MD trajectory times are given in Appendix A.

The models of NCP interacting with LANA_1-22_ peptide (LANA/NCP) or CENP-C_motif_ peptide from *R. norvegicus* (CENP-C/NCP) were derived from the corresponding X-ray structures (PDB IDs 1ZLA [24] and 4X23 [25], respectively). The LANA/NCP model contained *X. laevis* histones and *H. sapiens* α-satellite DNA sequence by default. To build a corresponding CENP-C/NCP model, the histone octamer in the original structure was replaced by an octamer containing *X. laevis* from an X-ray structure of a free NCP (PDB ID 1KX5 [73]). Unresolved DNA and histone fragments in the models were rebuilt using an alignment to our previously published models of the NCP [34]. Unresolved terminal LANA_1-22_ and CENP-C_motif_ residues were reconstructed manually using PyMOL version 2.5.5 (Schrodinger Inc., New York, NY, USA) [74], and in the case of CENP-C/NCP model, by additionally using the two-fold symmetry relation between the two CENP-C_motif_ peptides bound to the two symmetric sides of the NCP. Clashes in prepared models were repaired using FoldX version 3 [75]. For MD simulations, models with truncated histone tails were used; truncation sites were described previously in [34].

MD simulations were performed using previously published protocols [34]. In brief, GROMACS 2020.1 [76] with AMBER ff14SB force field [77] supplemented with parmbsc1 [78] DNA and CUFIX [79] ion parameter corrections were used. Model systems were placed in a truncated octahedron simulation box with periodic boundary conditions. For solvation, a TIP3P water model [80] was used, and Na and Cl ions were added to neutralize the charge and bring the ionic strength to 150 mM. Models were minimized (10,000 steps of steepest-descent gradient method) and equilibrated (in 5 steps: 4 steps with gradual reduction of restraining potential and 5th step of unrestrained 200-ps equilibration). Temperature was set to 300 K [81] and pressure was set to 1 bar [82]. MD production runs used 2 fs integration step and 1 ns frequency to output trajectory frames. Obtained trajectory times were 1000 ns and 400 ns for CENP-C_motif_ and LANA_1-22_ NCPs, respectively.

For trajectory analysis, custom scripts in Python 3 (see [34]), VMD version 1.9.3 (University of Illinois, Urbana-Champaign, IL, USA) [70] and PyMOL version 2.5.5 (Schrodinger Inc., New York, NY, USA) [74] (visualization), GROMACS 2020.1 [83] (trajectory preprocessing), and MDAnalysis version 2.6.1 [68] (coordinate manipulation, 3D alignment) were used. All trajectories were aligned in a nucleosome reference frame (NRF) which is based on the dyad symmetry axis and the superhelical axis (OY and OZ axes, respectively) [34]. Alignment was made by minimizing the root mean square deviation (RMSD) between the Cα-atoms of the histone folds α-helices (α1, α2, α3). As a reference trajectory of free NCP for comparison with peptide-bound NCPs, the 10-µs trajectory of NCP published in [34] was used.

Peptide–NCP atom–atom contacts were defined as pairs of non-hydrogen atoms at the distance of less than 4 Å. Residue–residue contacts between peptides and NCP were considered to be present if any of their atoms were in contact. Stable contacts were defined as residue–residue contacts present in at least 70% of MD frames across all simulation replicates (for CENP-C systems contacts of the two peptides contacting different sides of the NCP were treated).

The angle between α2-helices of H2A and H2B histones (α2-α2 angle) was defined as the angle between vectors connecting the first and last Cα-atoms of the H2A and H2B α2-helices following [36]. The distances between the α3-helices (α3-α3) or the αC-helices (αC-αC) of the two H2A molecules in NCP were defined as distances between the centers of mass of the Cα-atoms of the corresponding helices. The distributions (probability density functions) were visualized using a kernel density estimate with Gaussian kernels (SciPy realization).

### 4.3. Bioinformatics Analysis of Histone Variants

Amino acid sequences of the H2A histone family were collected from the HistoneDB database (https://histdb.intbio.org/ (accessed on 24 September 2023)) [6], extended with some newly characterized sequences, and manually curated. The Basic Local Alignment Search Tool (BLAST) was used to find sequences for Lilium, Giardia, and Microsporidians. The collected set contained 270 histone sequences of H2A histones with description that included species, class, and phylum of the organism.

Multiple alignments of amino acid sequences of various histone variants were constructed using the MUSCLE version 5.1 program [84] to analyze amino acid variations in the binding sites of the LANA peptide. The reference sequence (*X. laevis*) was derived from the X-ray structure (PDB IDs 1ZLA [24]).

Python scripts were written to find and analyze significant variations that potentially affected nucleosome stability and chromatin functioning [37]. TexShade version 1.26c was used to visualize the resulting alignments and important sequence cites [85].

### 4.4. Nucleosomal DNA Preparation

Non-fluorescently labeled nuclesomal DNA fragments were obtained as follows. pUC57 plasmids containing eight repeats of the high-affinity 147 bp Widom 603 sequence were amplified in *E. coli* XL-10 stain and purified via alkaline lysis (see details in Appendix A). The Widom 603 fragments were released via cleavage with EcoRV and separated from parent plasmid with polyethylene glycol precipitation in order to clear the vector backbone as described in [86]. Nucleosomal DNA fragments fluorescently labeled at the 35 and 112 bp positions were obtained following established protocols [44,87,88]. Briefly, DNA fragments (147 bp long) were amplified via PCR from the pUC57 plasmid containing the Widom 603 nucleosomal DNA sequence using the following primers: 

Forward 5’-ATCAGTTCGCGCGCCCACCTACCGTGTGAAGTCG[Cy3-dT]CACTCGG-3’ (where Cy3-dT is a nucleotide labeled with Cy3);

Reverse 5’-ATCCCAGGGACTTGAAGTAATAAGGACGGAGGGCC[Cy5-dT]CTTTCAACATCGAT-3’ (where Cy5-dT is a nucleotide labeled with Cy5). A Cleanup Standard kit (Evrogen, Moscow, Russia) was used to purify the amplified DNA fragments.

### 4.5. Histone Expression, Preparation of Histone Octamers

All four *H. sapiens* canonical histones were overexpressed in *E. coli* expression strains. *E. coli* BL21 (DE3) strain was used for histones H2B, H2A, and H3.1, and *E. coli* Rosetta 2 strain was used for histone H4 production. Each of the histones was overexpressed in the corresponding strain by growing on LB medium at 37 °C, 220 rpm to an optical density of 0.6–0.8 (OD_600_). Next, IPTG was added to the final concentration of 0.4 mM and bacterial growing continued at 37 °C, 220 rpm, for 4 h. Recombinant histones were purified, and histone octamer was assembled in strict accordance with the protocol described in [89]. Briefly, to prepare histone octamers, H2A, H2B, H3, and H4 histones were combined with molar ratio 1.5:1.5:1:1 in denaturing buffer (50 mM Tris pH 8.0, 7 M GuHCl, 10 mM DTT), dialyzed into refolding buffer (50 mM Tris pH 8.0, 2 M NaCl, 1 mM EDTA, 5 mM β-MeEtOH), and purified via gel filtration chromatography.

Denaturing SDS-PAGE with 18% gels was performed to analyze protein purity and content at the stages of overexpression and purification of single histones, as well as during the analysis of fractions after assembly and gel filtration of histone octamers.

PAGE with 4% gels was used to check the quality of nucleosomes (10 mM HEPES-Na pH 8.0, 0.2 mMEDTA, 10% glycerol). Similarly, 6% gels were used to analyze the specificity of binding of LANA_1-22_ to nucleosomes.

### 4.6. Nucleosome Assembly

NCPs were reconstituted by mixing 147 bp Widom 603 DNA and histone octamers at a 1:1.15 molar ratio. Dialysis was carried out at 4 °C, using buffers containing 10 mM Tris-HCl pH 8.0, 0.1% NP-40, 0.2 mM EDTA, 5 mM β-mercaptoethanol, and decreasing NaCl concentrations as described in [90]. If needed, nucleosomes were purified from nonspecific products by preparative electrophoresis in 4% PAAG as described in [44]. PAGE with 4% gels was used to check the quality of nucleosomes (10 mM HEPES-Na pH 8.0, 0.2 mM EDTA, 10%, glycerol). Fluorescently labeled nucleosomes were detected by in-gel FRET using a Typhoon scanner (GE Healthcare, Chicago, IL, USA) with excitation at 532 nm laser and emission at 670 nm (FRET between Cy3 and Cy5) or 580 nm (Cy3 signal) (Appendix A).

### 4.7. LANA_1-22_ and CENP-C_motif_ Peptides

LANA_1-22_ and CENP-C_motif_ peptides were synthesized by the Department of Molecular Basis of Neurosignalization, Shemyakin-Ovchinnikov Institute of bioorganic chemistry of RAS, Russia; sequences are described in Appendix A. LANA_1-22_ labels with 6-FAM at N-terminus were additionally synthesized for Kd measurements. 

### 4.8. Electrophoretic Assay for Peptide Binding

To assess the specificity of LANA_1-22_ binding to nucleosomes versus DNA, we used 6% PAGE (10 mM HEPES-Na pH 8.0, 0.2 mM EDTA, 10%, glycerol). LANA_1-22_ peptide at the concentration of 25 μM was mixed with Widom 603 DNA or nucleosomes at the concentration of 2 μM, and incubated for 30 min at +4 °C. Peptide–nucleosome complexes, intact nucleosomes and DNA in the gels were detected using Typhoon scanner (GE Healthcare, Chicago, IL, USA).

### 4.9. Fluorescence Polarization Studies

LANA_1-22_ binding to nucleosomes was quantitatively analyzed via a typical fluorescence polarization approach [91]. Briefly, FAM-labeled LANA_1-22_ peptide served as a tracer molecule and was incubated with varying concentrations of nucleosomes. Fraction of bound peptides was estimated based on the measured polarization anisotropy of the FAM label. Dissociation constants were estimated by fitting corresponding mathematical binding models.

Measurements were performed in nucleosome-binding buffer solution of the following composition: 10 mM TrisHCl pH 8.0, 200 μM EDTA, 5 mM 2-mercaptoethanol, 10 mM NaCl. FAM–LANA_1-22_ concentration was estimated with light adsorption of the FAM fluorophore and was kept constant at 100 nM in all measurements (OD at 495 nm was measured using Nanodrop 2000c Spectrophotometer, Thermo Scientific, Waltham, MA, USA, assumed molar extinction coefficient of FAM fluorophore was 75,000 M^−1^). Low adhesion microcentrifuge tubes (USA Scientific, Ocala, FL, USA) were used to mix nucleosomes with FAM–LANA_1-22_ peptides. Samples with various nucleosome concentrations were prepared by a two-fold serial dilution. Mixed samples were incubated for 30 min at +4 °C and then transferred to a microplate. Fluorescence polarization measurements were performed in black low-volume non-binding opaque 384-well microplates (Corning, Corning, NY, USA, cat. no. 3821BC) with BMG PolarStar Omega (BMG Labtech, Ortenberg, Germany) plate reader using 485 nm excitation and 520 nm emission filters. Every sample was distributed between five microplate wells, 40 µL of sample in each well. Similar protocols were used to prepare FAM–LANA_1-22_ peptides mixed with pure nucleosomal DNA. PMT gain and G-factors were automatically adjusted using 100 nM standard solution of fluorescein sodium salt (Merck, Rahway, NJ, USA, cat. No. 46970) in nucleosome-binding buffer. All measurements were repeated 10 times.

A Python script was written to analyze the raw data (i.e., parallel and perpendicular components of the fluorescence intensity measurements), implementing recommendations of [92]. Polarization anisotropy values were calculated considering (1) background signal correction due to scattering of light in pure buffer solutions, and (2) potential fluorescence quenching estimated by the Q-factor (Q is the ratio of fluorescence intensities of bound and free species measured under the same experimental conditions).

LMFIT version 1.2.2 python library was used for a nonlinear regression fit of the measured polarization anisotropy data to the binding model equation [93]. A simple direct binding model was used [92] (with approximation that nucleosomes have identical, non-interacting binding sites), see Equation (1). In this model, binding constant (Kd) and anisotropy of completely bound species were subject to fitting and statistical estimation.
(1)$$A=\frac{QFA_{bound}+(1-F)A_{free}}{1-(1-Q)F}\;\;\;\;F=\frac{K_{d}+{L_{0}^{*}}_{}^{}+nP_{0}+\sqrt{({K_{d}+{L_{0}^{*}}_{}^{}+nP_{0})}^{2}-4nP_{0}{L_{0}^{*}}_{}^{}}}{2{L_{0}^{*}}_{}^{}}$$

In Equation (1), ${L_{0}^{*}}_{}^{}$ is a concentration of the LANA_1-22_; $P_{0}$ is the concentration of the nucleosomes; Q is a Q-factor; $A_{bound}$ and $A_{free}$ are the anisotropies of the completely free and bound species, respectively; and n is a number of binding sites on the nucleosomes.

### 4.10. Single-Molecule FRET Analysis of Nucleosome–Peptide Interactions

Single-particle FRET measurements were conducted with LSM710-Confocor3 laser scanning confocal microscope (Carl Zeiss, Jena, Germany) as described previously [43]. Nucleosomes at a concentration of 1 nM with peptides at a concentration of 10 μM were incubated for 15 min at +4 °C in a buffer containing 17 mM HEPES pH 7.6, 2 mM Tris-HCl, 0.8 mM Na_3_EDTA, 0.11 mM 2-mercaptoethanol, 11 mM NaCl, 1.1% glycerin, and 12% sucrose. Samples were subjected to spFRET measurements in a multi-well silicon chamber (Ibidi GmbH, Gräfelfing, Germany) fixed on a cover glass. Datasets were converted to photon HDF5 format. To evaluate the level of background signal and reveal fluorescence bursts, spFRET data were analyzed using the FretBursts version 0.7.1 software [94]. The bursts searching window was set to 10 successive photons. The conditions of burst acceptance were as follows: the burst should have more than 40 photons and their maximum brightness should be at least 5 times higher than the average background signal level. Eventually, each measurement included 800–1400 particles and was characterized by relative frequency distribution of proximity ratio E_PR_ as described in [94]. Such profiles were approximated as one Gaussian curve using LMFIT.

## 5. Conclusions

Acidic patch-binding peptides present perspective compounds that can be used to modulate chromatin functioning. In this study, through a combination of bioinformatics, computational, and experimental methods, we have characterized their binding to nucleosomes in terms of intermolecular interaction patterns, binding strength, specificity, and effects on the stability and dynamics of nucleosomes. Our findings suggest that sequences of such peptides and their conformation in the bound state may be variable. The binding efficiency of such peptides, in the first place, depends on their ability to form interactions with the negatively charged amino acid residues of the acidic patch through arginies, lysines, and in part through hydrogen bonds with serines. However, additional interactions are important for the overall stability of the peptide–nucleosome complex. These interactions may be formed by other residues of the acidic patch region (e.g., H2BQ44, H2BL103, H2BH106, H2AY57), or peptides may engage other parts of the nucleosomal surface (e.g., the region formed by H3 and H4 histones around the nucleosomal pore). Taken together, these observations suggest that the design of de novo high-affinity artificial peptide binders that would engage a vast spectrum of various interactions with the nucleosome should be possible.

Our analysis of nucleosome dynamics suggests that peptide binding, on the one hand, leads to the overall stabilization of the nucleosomes, and, on the other hand, may introduce small changes in the geometry and conformation of the histone octamer. The latter changes alter the plasticity of the nucleosome core and may affect allosteric communication in the nucleosome.

## Figures and Tables

**Figure 1 ijms-24-15194-f001:**
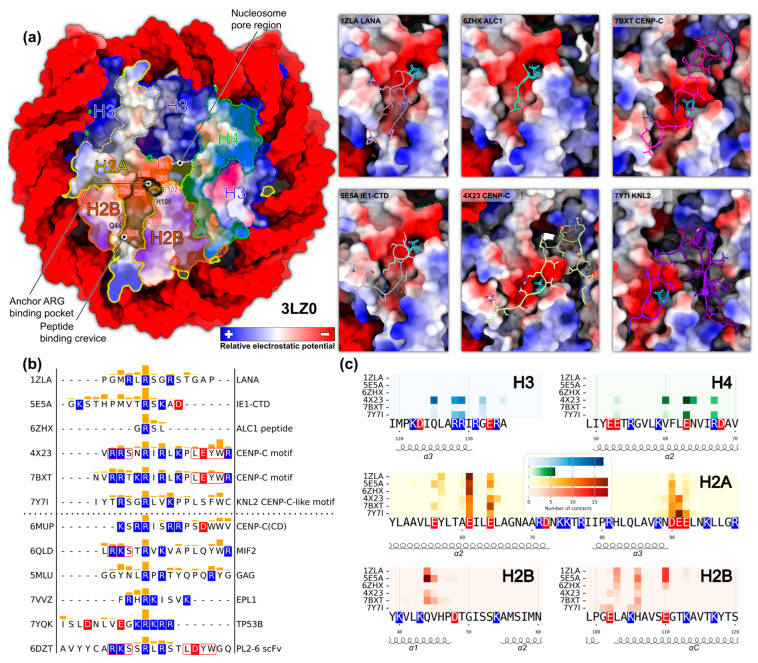
Analysis of interactions between acidic patch-binding peptides and nucleosomes from the PDB database. (**a**) Left—structure of a whole nucleosome core particle (PDB ID 3LZ0) in the surface representation showing the typical binding site of peptides: the anchor arginine pocket and the crevice along the H2A-H2B dimer. Right—zoomed-up regions showing different peptides interacting with the acidic patch in various PDB structures. The surface is colored according to electrostatic charge. Anchor arginine is highlighted. (**b**) Sequences of peptides (above the dotted line) and select protein sequence motifs (below the dotted line) interacting with the acidic patch aligned by the position of the anchor arginine in the center. Only residues resolved in PDB structures are shown. The orange bars show the relative number of atom–atom contacts. Red rectangles highlight similar sequence motifs in different structures that surprisingly interact in some structures with the nucleosome but do not interact in others, highlighting the high structural polymorphism of interactions. Positive and negative residues are highlighted by red and blue colors, respectively. (**c**) Diagrams showing contacting residues of the histones with the acidic patch-binding peptides in different structures.

**Figure 2 ijms-24-15194-f002:**
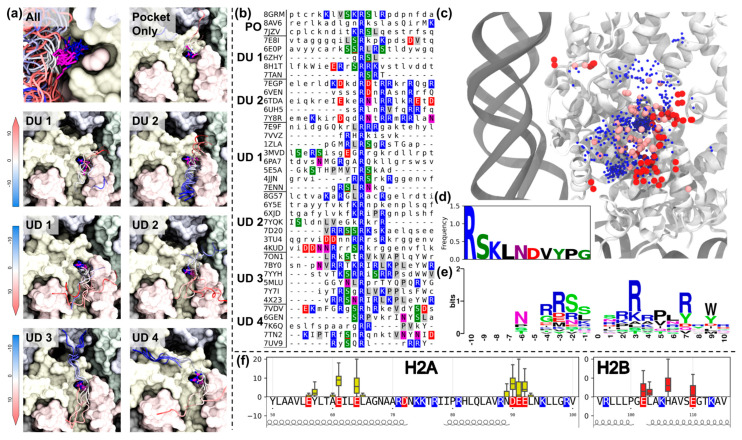
Comprehensive analysis of acidic patch interactions in structures of nucleosomes bound to proteins or peptides from the PDB database. (**a**) Different conformational arrangements of acidic patch-binding motifs grouped by type. (**b**) Alignment of protein motifs with respect to the anchor arginine grouped by their conformation arrangement in the structure. Uppercase letters denote amino acids in contact with the nucleosome, they are colored according to their type or identity (positive in blue, negative in red, hydrophobic in grey, serines in green, and asparagines in purple). Underlined serine residues are involved in hydrogen bonding with glutamate side chains. (**c**) Location of positively charged (blue), negatively charged (red), and serine residues of nucleosome-binding protein motifs around the acidic patch. (**d**) The frequency of different amino acid residues in the acidic patch-binding protein motifs in the vicinity of the anchor arginine (±5 residues, excluding the anchor arginine itself). (**e**) A sequence logo plot for the alignment shown in panel (**b**). (**f**) Distribution of contacts between histone residues and acidic patch-binding motifs in the analyzed structures.

**Figure 3 ijms-24-15194-f003:**
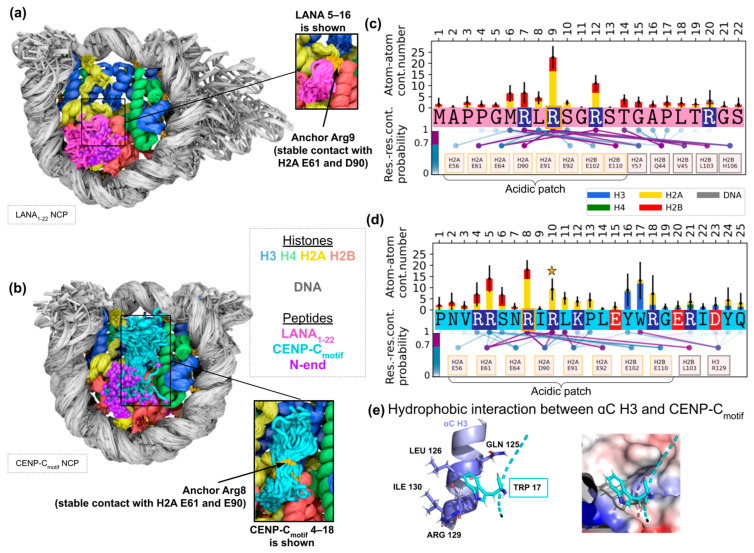
MD simulations of NCP with LANA_1-22_ and CENP-C_motif_. (**a**,**b**) Overview of MD simulations of NCP with LANA_1-22_ (**a**) and CENP-C_motif_ (**b**). Overlay of MD snapshots spaced 10 ns apart. The dynamics of anchor arginine residue and neighboring parts of the peptides are shown as insets. (**c**,**d**) Contact profiles of LANA_1-22_ (**c**) and CENP-C_motif_ (**d**) with the NCP. The trajectory average (over all trajectories) number of atom–atom contacts between peptide and NCP is shown on top of the sequence. Residue–residue contacts between the peptide and acidic patch of the nucleosome as well as other stable residue–residue contacts are shown as lines below the sequences. Line color intensity is in a direct ratio with contact frequency. Purple lines show stable contacts. Anchor arginines are highlighted with orange frames. An orange asterisk marks CENP-C_motif_ R10, which can partially substitute R8 in contacting the negatively charged binding pocket of R18 (see text and Appendix A). (**e**) Atomistic details of stable hydrophobic interaction of CENP-C_motif_ with H3 α3-helix. The electrostatic surface was calculated and visualized using PyMOL with the APBS plugin.

**Figure 4 ijms-24-15194-f004:**
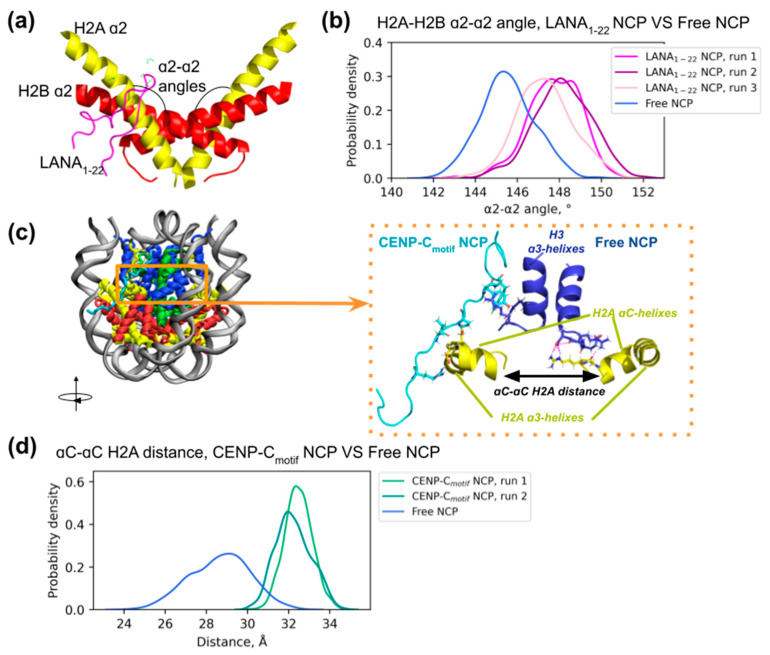
Effects of CENP-C_motif_ and LANA_1-22_ binding on NCP geometry. (**a**,**b**) Effects of LANA_1-22_ binding on H2A-H2B dimer geometry within the NCP. (**a**) Illustration of the angle between the H2A α2 and H2B α2-helices (as a measure of dimer bending). (**b**) α2-α2 angle distributions in MD simulations of free NCP and LANA_1-22_ NCP. (**c**,**d**) Alteration of NCP central region upon CENP-C_motif_ binding. (**c**) Interaction of CENP-C_motif_ peptide with α3-αC H2A helices and α3 H3 helix and its effect on reorganization of H2A-H3 inner-contacts. (**d**) Distance between αC-helices of two H2A copies in the NCP (as shown in panel (**c**)). Comparison of the MD simulations with and without CENP-C_motif_ peptide. The distance was defined as the distance between the centers of mass of the helices’ Cα-atom.

**Figure 5 ijms-24-15194-f005:**
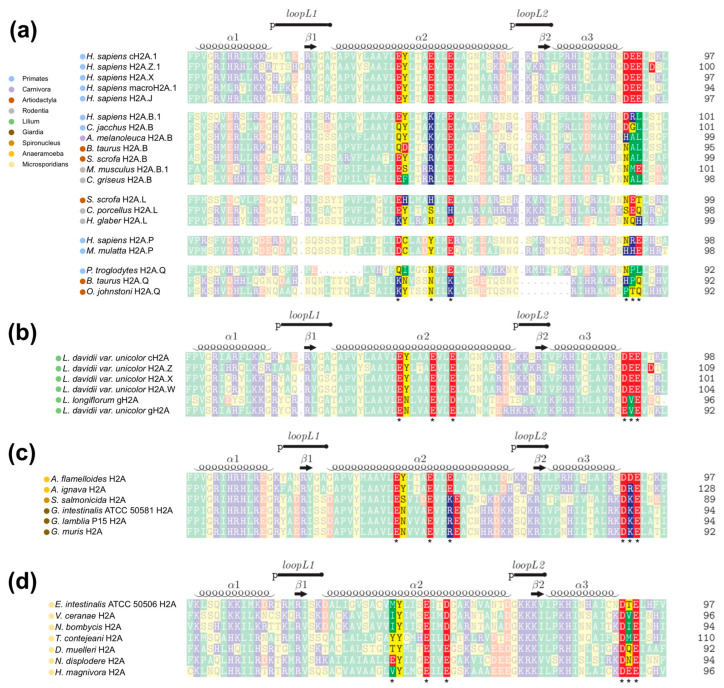
Multiple sequence alignments of representative sequences of H2A variants in animals (**a**), lilium (**b**), Metamonada (**c**) and microsporidians (**d**). Highlighted amino acids are the important contact sites on the histone octamer surface engaged during peptide binding. The color of the amino acids correspond to their hydropathicity: red—acidic, blue—basic, yellow—polar uncharged, green—hydrophobic nonpolar. Acidic patch amino acids are indicated by asterisks. The color of the dot located to the left of the sequences specifies different phyletic groups.

**Figure 6 ijms-24-15194-f006:**
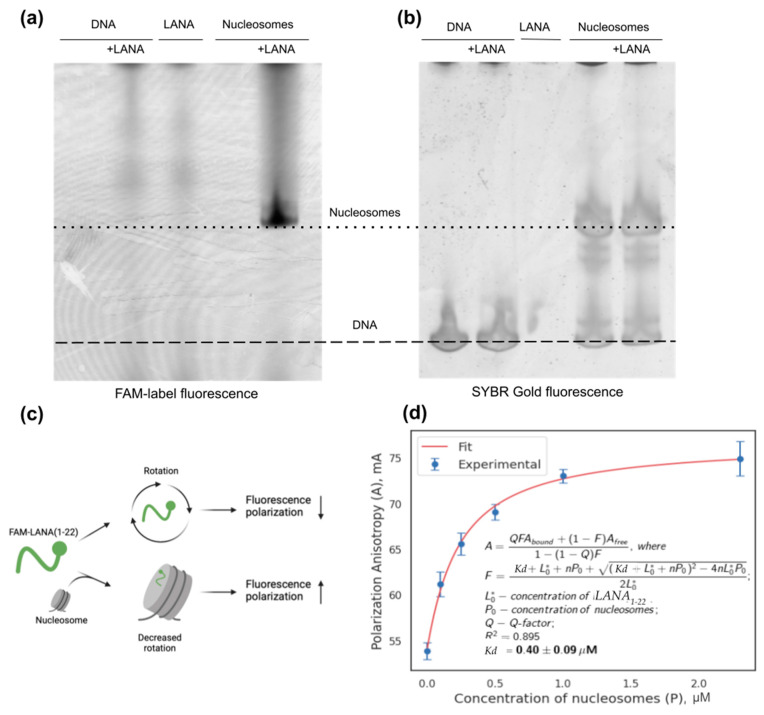
Estimating LANA_1-22_ specificity and dissociation constant (Kd) of nucleosome binding. (**a**,**b**) Analysis of LANA_1-22_ binding to DNA and nucleosomes via electrophoretic assay. Nucleosomal DNA, nucleosomes, DNA incubated with LANA_1-22_, and nucleosomes incubated with LANA_1-22_ were subject to 6% PAGE. The same gel was scanned using a Typhoon imager in different channels to detect the presence of FAM-labeled LANA_1-22_ (panel (**a**)) and DNA/nucleosomes via SYBR Gold fluorescence (panel (**b**)). Bands corresponding to nucleosomes and free DNA are indicated with dotted and dashed lines, respectively. (**c**) Schematic diagram of the experimental fluorescence polarization assay. Nucleosomes at varying concentrations were mixed with 100 nM of FAM-labeled LANA_1-22_ and incubated for 30 min at +4 °C. FP measurements were performed. The binding of LANA_1-22_ with nucleosomes was measured through the increase of polarization anisotropy due to the decreased rotational diffusion of the FAM-labeled LANA_1-22_. (**d**) Dependence of polarization anisotropy on the concentration of nucleosomes. Average experimentally obtained values are plotted together with their uncertainty estimates (SEM values). Measurements were repeated five times at each concentration. The solid line represents the fit of a mathematical model shown in the inset of the plot. The estimates of Kd and other fit parameters are shown in the plot.

**Figure 7 ijms-24-15194-f007:**
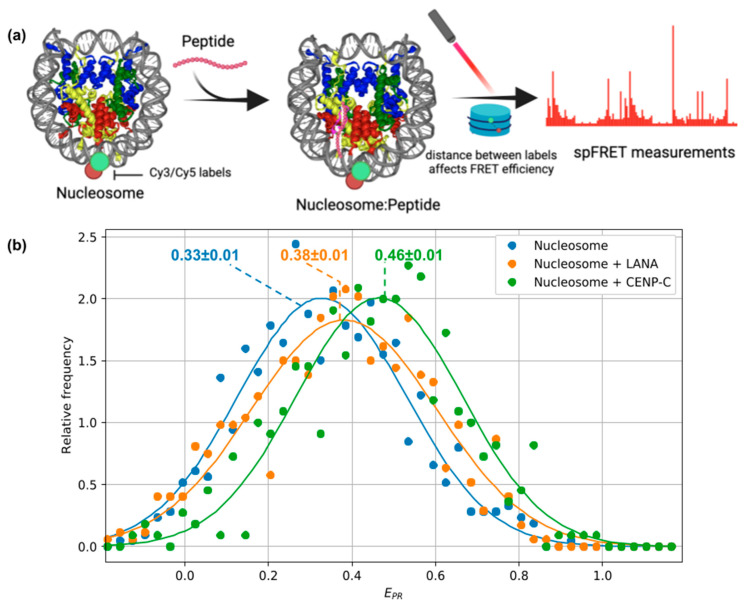
Peptides with high affinity to the acidic patches on nucleosomes affect nucleosome structure as revealed by spFRET measurements. (**a**) Schematic diagram of the experimental spFRET assay. Nucleosomes at a concentration of 1 nM with peptides at a concentration of 10 μM were incubated for 15 min at +4 °C. Such samples were subjected to spFRET measurements. (**b**) Frequency distributions of the proximity ratios (E_PR_) characterizing FRET in fluorescently labeled nucleosomes in the absence or in the presence of LANA_1-22_ or CENP-C_motif_.

## Data Availability

MD simulation trajectories, protocols, and an interactive viewer of nucleosomes interacting with the acidic patch-binding proteins and peptides are available for preview and download from GitHub at https://intbio.org/Oleinikov_et_al_2023/ (accessed on 24 September 2023).

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
