# Peer review of "Interactions of Nucleosomes with Acidic Patch-Binding Peptides: A Combined Structural Bioinformatics, Molecular Modeling, Fluorescence Polarization, and Single-Molecule FRET Study"

_ijms, 2023, doi:10.3390/ijms242015194_

Round 1

Reviewer 1 Report

This work studied the interaction patterns of typical acidic patch binding peptides and nucleosomes by simulation and experiments. Hypotheses of binding theory were also discussed. It offers better understanding of the interaction of nucleosomes with acidic patch binding motifs. While a lot of data and results were discussed in the article, overall it can be more concise. Besides here are some minor suggestions:

In Figure 1a, it is mentioned that the surface is colored according to electrostatic charge. To make it easier to understand, please add the color legend to the figure.

Page 6. Line 227 is referring to the wrong figure (figure 1a instead of figure 2a).

The “LANA” bands in Figure 6a and 6b were annotated as “gap” in the supplementary original images. If the label is wrong, please correct it. If not, please explain the meaning of “gap”.

Author Response

We would like to thank the referee for the positive evaluation of our manuscript and for the improvement suggestions. Below we provide the answers to the referee's suggestions.

  1. Referee's comment: While a lot of data and results were discussed in the article, overall it can be more concise.
    Answer: we tried to make section 2.3 more concise. However, we opt not to significantly edit the manuscript, since there is no consensus between the referee's (referee 3 asked to expand the introduction section).

2.  Referee's comment: In Figure 1a, it is mentioned that the surface is colored according to electrostatic charge. To make it easier to understand, please add the color legend to the figure.

Answer: the color legend was added.

3. Referee's comment: Page 6. Line 227 is referring to the wrong figure (figure 1a instead of figure 2a).

Answer: this was fixed.

4. Referee's comment: The “LANA” bands in Figure 6a and 6b were annotated as “gap” in the supplementary original images. If the label is wrong, please correct it. If not, please explain the meaning of “gap”.

Answer: the lanes labeled with the "gap" are the empty lanes of the gel. We change the label to the word "empty" 

Respectfully yours,

on behalf of co-authors,

Dr. Alexey K. Shaytan

Reviewer 2 Report

SUMMARY:

Using computational and experimental methods, Oleinikov et al. characterized the interactions patterns and dynamical effects of nucleosomes with their acidic patch binding peptides.

Initially, they conducted a comprehensive bioinformatic analysis of the available structures of nucleosome-peptide complexes in PDB database, and confirmed that the interaction of an anchor arginine residue of a peptide with a negatively charged pocket on the surface of the octamer is a dominant feature.

Subsequently, molecular dynamics simulations were employed to explore the binding of LANA1-22 and CENP-C to nucleosomes. These simulations unveiled the critical role of transient atom-atom interactions in maintaining the overall stability of the nucleosome-peptide complex. Additionally, subtle yet discernible alterations in the histone core's geometry were observed upon peptide binding.

To validate these findings, Oleinikov and colleagues utilized fluorescent polarization assays and biochemical assays. Their results showed that LANA1-22 binds to nucleosomes at sub-micromolar concentrations, and this binding is specific to nucleosomes rather than DNA. Furthermore, to gain insights into the impact of peptide binding on nucleosome stability and conformation, they employed single-molecule FRET technique. These experiments demonstrated that binding of LANA1-22 and CENP-C peptides generally leads to nucleosome compaction.

Overall, this study offers valuable insights into diverse interaction patterns between nucleosomes and peptides, which can be instrumental in the development of novel peptides. Nevertheless, before publication, the authors need to address and clarify several concerns listed below.

Major comments:

1.     In Figure 6a, lanes 2 (LANA+DNA) and 3 (LANA only) display smeared bands. While the authors have likely assessed the quality of the synthesized LANA peptide, it is puzzling that the LANA1-22 peptide itself exhibits smeared bands on the gel. This may also be a contributing factor to the pronounced smearing observed in lane 5 (Nucleosome+LANA). It is necessary to demonstrate the quality of the LANA1-22 peptide.

2.     In Figure S6.3, the nucleosome position is specified to be within the range of 400 bp to 500 bp. However, in Figure 6b and the original gel for Figure 6b, the nucleosome position indicated by the dashed line appears to fall within the 500 bp to 600 bp range. The authors should address this inconsistency for clarification.

3.     By means of MD simulations, the authors showed that the removal of the terminal segments of the LANA1-22 peptide results in transient interactions with the NCP, resulting in peptide destabilization. It would be beneficial to substantiate this computational outcome through biochemical assays or single-molecule FRET experiments.

Minor comments:

1.     In Figure 4b, the label for the X-axis should be "angle." The same issue is observed in Supplementary Figure S4.2. Please make the correction accordingly.

2.     Please label the protein SDS page gel shown in Supplementary Figure S6.2, and provide an explanation for the dashed and solid lines.

3.     The annotations for the lanes on the original gel image files are not accurate. For example, for Figure 6a, Lane 5 should be labeled as “LANA” but not “gap”. Please make the corrections.

Reviewer 3 Report

The authors presents a comprehensive study on the interactions of nucleosomes with acidic patch binding peptides. The manuscript also identified some common binding patterns and interactions for peptides-nucleosome complex stabilization. The study also performed experimental validation on LANA, and confirmed their findings mirrors previous knowledge and with some new insights as well. Overall, the paper is well-written. It also provides a comprehensive and integrative analysis, and the findings have significant implications for understanding chromatin dynamics and potential therapeutic applications. Here are some minior points which might be beneficial for improvement:

1. The expansion of introduction on recent related studies in interactions of nucleosomes with acidic patch binding peptides might be beneficial for audience to better understanding the recent advances in this topic.

2. The authors can incorporate the potential therapeutic applications of the results of current in the abstract for presenting a larger picture to audience.

Author Response

We thank the referee for a thorough evaluation of our manuscript and for the suggestions. Below is our response:

  1. Referee's comment: The expansion of introduction on recent related studies in interactions of nucleosomes with acidic patch binding peptides might be beneficial for audience to better understanding the recent advances in this topic.

Answer: a following paragraph was added to the introduction.

"Fang et al. have studied the interactions of the chromatin-tethering domain of IE1 from human cytomegalovirus (hCMV) with the nucleosome core particles [26]. They have found that similar to LANA1-23 this peptide impairs the compaction of higher-order chromatin structure, suggesting that in vivo IE1 loosens up the folding of host chromatin during hCMV infections. Kato et al. have studied the interactions of CENP-Cmotif peptides with centromeric nucleosomes [25]. They have shown that CENP-Cmotif peptides also bind to canonical nucleosomes, but with lower affinity. The higher affinity of these peptides to centromeric nucleosomes come from an additional interaction with the hydrophobic C-terminal tail of centromeric H3 histone variants. Teles et al. have recently explored the effects of four nucleosome binding peptides (LANA1-23, HMGN2, H4 histone tail and an artificial peptide GMIP1) both in vitro and in vivo [27]. Their study suggested that H4 histone tail derived peptide stands out, is highly specific for targeting the nucleosome, with important effects on the final nucleosome structure and robust in vivo effects. However, additional experiments are needed to understand if these effects are solely due to acidic patch binding or other nonspecific interactions of the peptide. Previous studies by Chodaparambil et al. suggested that H4 tail has multiple modes of interaction with the nucleosome including non-specific ones [28]."

2. Referee's comment: The authors can incorporate the potential therapeutic applications of the results of current in the abstract for presenting a larger picture to audience.

Answer: the following sentence was modified in the abstract

"Acidic patch binding peptides present perspective compounds that can be used to modulate chromatin functioning by disrupting interactions of nucleosomes with natural proteins or alternatively targeting artificial moieties to the nucleosomes, which may be beneficial for the development of new therapeutics. "

Respectfully yours,

Dr. Alexey K. Shaytan

Round 2

Reviewer 2 Report

The authors have addressed my previous concerns. I think this study will be a nice contribution to the field and thus I am supportive of publication.